# Antioxidant and Antimicrobial Peptides Derived from Food Proteins

**DOI:** 10.3390/molecules27041343

**Published:** 2022-02-16

**Authors:** Guadalupe López-García, Octavio Dublan-García, Daniel Arizmendi-Cotero, Leobardo Manuel Gómez Oliván

**Affiliations:** 1Food and Environmental Toxicology Laboratory, Chemistry Faculty, Universidad Autónoma del Estado de México, Paseo Colón Intersección Paseo Tollocan s/n. Col. Residencial Colón, Toluca 50120, Mexico; logg_sf@hotmail.com (G.L.-G.); lgolivan74@gmail.com (L.M.G.O.); 2Department of Industrial Engineering, Engineering Faculty, Campus Toluca, Universidad Tecnológica de México (UNITEC), Estado de México, Toluca 50160, Mexico; arcoda21@gmail.com

**Keywords:** bioactive peptides, antioxidant capacity, antimicrobial capacity, proteins

## Abstract

Recently, the demand for food proteins in the market has increased due to a rise in degenerative illnesses that are associated with the excessive production of free radicals and the unwanted side effects of various drugs, for which researchers have suggested diets rich in bioactive compounds. Some of the functional compounds present in foods are antioxidant and antimicrobial peptides, which are used to produce foods that promote health and to reduce the consumption of antibiotics. These peptides have been obtained from various sources of proteins, such as foods and agri-food by-products, via enzymatic hydrolysis and microbial fermentation. Peptides with antioxidant properties exert effective metal ion (Fe^2+^/Cu^2+^) chelating activity and lipid peroxidation inhibition, which may lead to notably beneficial effects in promoting human health and food processing. Antimicrobial peptides are small oligo-peptides generally containing from 10 to 100 amino acids, with a net positive charge and an amphipathic structure; they are the most important components of the antibacterial defense of organisms at almost all levels of life—bacteria, fungi, plants, amphibians, insects, birds and mammals—and have been suggested as natural compounds that neutralize the toxicity of reactive oxygen species generated by antibiotics and the stress generated by various exogenous sources. This review discusses what antioxidant and antimicrobial peptides are, their source, production, some bioinformatics tools used for their obtainment, emerging technologies, and health benefits.

## 1. Introduction

Human beings require oxygen (O_2_) to produce energy via the oxidative metabolism of the mitochondria; they produce ATP, which is the molecule from which the energy necessary for various vital processes is released. However, an excess of O_2_ in cells is harmful due to the formation of reactive species generated during oxidation. The mitochondria are considered to be the main endogenous source of free radicals (FRs) since, in order to produce ATP, the oxidative metabolism reduces 95–98% of the oxygen in a tetra-electronic manner; the rest is reduced in a monoelectronic manner, forming intermediaries known as reactive oxygen species (ROS) or reactive nitrogen species (RNS). These are normally produced within the body and act as a defense against infections by bacteria and viruses, participate in the maturation of the reticulocytes, and participate in the degradation of proteins. Each mitochondria produces approximately 7–10 M FR/day and is capable of producing other reactive species of biological relevance that are formed because of a cascade reaction, such as nitric oxide (NO), peroxynitrite (ONOO-), and hydrogen peroxide (H_2_O_2_) [1].

Oxidative stress is responsible for cellular degeneration because FRs can chemically react with proteins, lipids, and DNA, thereby producing various modifications, cell damage, and even cell death. It has been reported that FRs oxidize the amino acids of proteins, leading to the formation of carbonyl groups, the association of protein fragments via the cross-linking of disulfide bonds, the breaking of peptide bonds, the loss of affinity for metals, and an increase in hydrophobicity, resulting in proteins with oxidative damage that present deteriorations in hormonal and enzymatic activity, as well as ion transport, and a greater susceptibility to proteolytic degradation [2]. Polyunsaturated fatty acids (PUFAs), once oxidized by FRs, lead to lipoperoxidation, a reaction in which PUFAs give their electrons to FRs and generate changes in the molecular structure of the membrane, as well as the formation of disulfide bonds between membrane proteins, that leads to a loss in permeability and the stability of the membrane, resulting in cell death [3].

The free radicals also attack DNA, damaging the genes that codify proteins, which leads to a lack of execution in cell functions. It has been reported that OH· is responsible for various modifications in deoxyribose resulting in the release of nitrogenated bases (which are linked together by the aforementioned sugar) and the breaking of one or both chains, resulting in deletions, mutations, chromosomic rearrangements, and the activation or inactivation of genes, all of which affect the biosynthesis of DNA chains; likewise, reactive oxygen species produce errors during the transcription and translation of RNA [4,5].

To counter the harmful effects of O_2_ and its derivatives, the cell possesses enzymes, electron scavengers and nutrients that act as an antioxidant system in charge of maintaining the equilibrium of oxidation–reduction reactions and cell survival. However, FRs are also generated by exogenous sources and factors that favor their formation, such as exposure to X-rays, ozone, tobacco, air contaminants, and industrial products; certain medications and inadequate eating habits; the consumption of foods with low nutritional quality and antioxidant capacity, fast food with high fat content, junk food, canned goods that contain additives, drinks with high sugar content; and the low consumption of natural foods, which provokes an increase in the concentration of FRs and a loss of equilibrium between the speed of formation and their neutralization by the body’s endogenous antioxidant system. This leads to oxidative stress, which has been linked to obesity, chronic degenerative illnesses, neurodegenerative illnesses, cardiovascular illnesses, diabetes mellitus and cancer, as well as severe cell damage [6].

The harmful effects of oxidative stress on human health can be reduced via the ingestion of dietary antioxidants present in various foods. A balance must be achieved between the abundance of ROS and antioxidants, which are compounds capable of donating electrons, to stabilize the free radicals and neutralize their harmful effects. These can be of endogenous origin (synthesized within the body), and superoxide dismutase and glutathione peroxidase are among the endogenous antioxidants [7]).

Exogenous antioxidants (from external sources), such as coenzyme Q and bioactive peptides (antioxidants and antimicrobials), that humans can obtain from their diet can act in two ways: (1) preventing the excessive production of free radicals, thus avoiding cell damage due to the effects of oxidative stress; or (2) controlling the levels of free radicals after damage has been produced, thus preventing further damage and thereby preventing some symptoms of illnesses produced by the effects of oxidative stress [7,8].

Many studies have suggested that some side effects produced by various antibiotics such as amoxicillin, dapsone, vancomycin, ampicillin, doxorubicin, isoniazid, rifampin, ciprofloxacin, enrofloxacin, chloramphenicol, and gentamicin can increase oxidative stress in human cells [9,10,11,12,13,14,15,16]. This has motivated the search for new molecules with antimicrobial activity that are effective with fewer side effects. Antimicrobial peptides are characterized by being positively charged molecules and presenting hydrophobic residues, which can act against a wide spectrum of microorganisms including Gram-positive bacteria and Gram-negative bacteria, viruses and fungi; accordingly, these peptides can be used to contribute adjuvants to medication and reduce the side effects of these drugs [7].

The present review focuses on what antioxidant and antimicrobial peptides are, their source, production, some bioinformatics tools for their obtainment, emerging technologies and health benefits.

## 2. Bioactive Peptides

Bioactive peptides are short sequences of 2–40 units of amino acids that are inactive within the precursor protein and, once released via chemical or enzymatic hydrolysis, in vivo digestion, or food processing can carry out a variety of biological activities [17,18]. It has been reported that bioactive peptides exhibit hormonal activity or activity like that of drugs and can be classified according to their mechanism of action as antioxidants, antimicrobial agents, antithrombotic agents, antihypertensive agents, opioids, immunomodulators or metal chelators [19,20,21,22,23]. Several studies have indicated that these activities are determined by the degree of hydrolysis (number of peptide bonds broken in relation to the original protein), substrate concentration, enzyme/substrate ratio, incubation time, microorganism employed, and physicochemical conditions such as pH and temperature [24].

### 2.1. Sources of Bioactive Peptides

Peptides have been identified and isolated from animal and vegetable sources, and they are abundantly found in protein hydrolysates [23] and fermented dairy products, as well as some fungi such as *Ganoderma lucidum* [25]. They can be obtained via protein digestion by employing endogenous or exogenous enzymes, microbial fermentation, processing, or gastrointestinal digestion. Due to the relevance of these emerging molecules and with the aim of reducing costs, other protein sources have been sought; among these are food industry by-products, such as the heads and viscera of fish and bird feathers [26,27,28]. Some studies have reported that, depending on the protein source, the enzyme used, and the processing conditions, obtained biological activity and the peptides vary [21,29,30], as shown in Table 1.

#### 2.1.1. Peptides of Animal Origin

Red meats, birds, fish and eggs are valuable sources of protein that are not present in other foods. Bioactive substances, such as fatty acids (conjugated linoleic acid), carnosine, L-carnitine, glutathione, taurine, coenzyme Q10, and creatin, have also been identified in proteins of animal origin [31], as have peptides that have been reported to be physiologically functional.

Livestock, sheep, goats, pigs and poultry are the most common sources of meat for human consumption and proteins. From pork, for example, RPR, KAPVA and PTPVP—peptides with antihypertensive activity—have been identified [32]. Additionally, the most-studied peptides are those derived from milk caseins, such as lactotransferrin, which possesses antimicrobial activity towards a wide range of microorganisms including *Staphylococcus* spp. and *Streptococcus pyogenes* [33].

Another source of bioactive peptides is seafood, such as algae, fish, mollusks and crustaceans, from which some endogenous bioactive peptides, antioxidants and angiotensin-converting enzyme inhibitors have been identified, through the fermentation of salmon, sardine, anchovy, carp and mackerel [27,30,34,35,36]. Table 1 shows some sources of bioactive peptides of animal origin.

#### 2.1.2. Peptides of Vegetable Origin

The procurement of peptides from proteins of vegetable origin through enzymes of animal, vegetable, or microbial origin has also been reported. Some protein sources of vegetable origin from which bioactive peptides have been isolated are soy milk, soy seed, wheat, corn, rice, barley, and sunflower; it has been found that these present antihypertensive, hypercholesteremic, anti-obesity and anticarcinogenic effects [37]. Likewise, it has been reported that 51% of the peptides present in plants exhibit antimicrobial activity and a structure similar to those of animals and insects [38,39]. In Table 1, some sources of bioactive peptides of vegetable origin can be seen.

#### 2.1.3. Fungi-Derived Peptides

Fungi are one of the most important natural sources of bioactive compounds due to the presence of numerous products with therapeutic properties. Commonly used in traditional Chinese medicine for centuries, fungi are rich in natural antioxidants, antimicrobials, antitumor agents, antiviral agents, and immunomodulatory agents, with medicinal effects proven by researchers [1,2,3]. It has been shown that fungi can produce many bioactive compounds such as polysaccharides, phenols, sterols, proteins. and peptides. Which are responsible for the therapeutic effects attributed to this species [52,53,54,55,56].

Fungi such as *Ganoderma lucidum* (Curtis: Fr.) Karsten (Ling-chih or reishi), *Ganoderma tsugae, G. lucidum,* and *Marasmius oreades* were analyzed by Sun et al. (2004) [25], who isolated a fraction of bioactive peptides called GLP (peptide from *G. lucidum*) [25]. This peptide fraction has a molecular mass <10,000 Da, while the N-terminal sequence is unknown. The GLP fraction showed a high antioxidant activity comparable to that of the synthetic antioxidant butylhydroxytoluene (BHT) in soybean oil and lard systems. The authors showed that the GLP fraction at 0.1% (w:w) after 13 days inhibited the oxidation of soybean oil and lard at rates of 60 and 30%, respectively. The GLP fraction blocked the oxidation of polyunsaturated fatty acids responsible for the formation of lipid hydroperoxides by inhibiting 90% of lipoxygenase activity in vitro at 0.3 mg/mL. Furthermore, using mouse liver homogenates, 0.7 mg/mL of GLP fraction inhibited 72% of H_2_O_2_-induced malondialdehyde formation resulting from the lipid peroxidation of polyunsaturated fatty acids.

Finally, although the antioxidant properties of *G. lucidum* are mainly attributed to the fraction of water-soluble polysaccharides, the authors showed that aqueous extracts of low molecular weight, mostly made up of peptides, showed greater antioxidant activity than fractions of *G. lucidum*, which has high molecular weight [25]. A bioactive peptide called plectasin extracted from *P. nigrella* mRNA (FGCNGPWDE DDMQCHNHCK SIKGYKGGYC AKGGFVCKCY) was identified [57,58]. With a molecular mass of 4398.80 Da, plectasin is considered a bactericide, with antimicrobial activity against Gram-positive bacteria tested at physiological ionic strength in contrast to most vertebrate defensins, which require very low ionic strength. On the other hand, this peptide is selective for bacterial cells over mammalian cells in vitro, as it has not shown cytotoxicity against murine L929 fibroblasts or normal human epidermal cells. In an in vivo study using mouse models of pneumococcal peritoneal infection, the anti-infective action of plectasin was highlighted [58].

Due to its therapeutic potential, plectasin is considered to be a novel antimicrobial peptide with excellent penetration into cerebrospinal fluid, suggesting its possible efficacy in the treatment of central nervous system infections caused by Gram-positive pathogens, such as pneumococcal meningitis.

#### 2.1.4. Peptides from Agri-Food By-Products

Recent innovations for the generation of peptides with biological activity include the use of proteins from food by-products such as substrates for enzymatic and microbial action to take advantage of their technological properties, improve nutritional properties, reduce production costs. The by-products used as protein sources include bones from the meat industry, from which 10–13% of collagen and gelatin, which are proteins that contains biologically active peptides within their sequences, can be obtained [59]. The bones of hydrolyzed marine by-products, such as tuna bones, have been found to contain the VKAGFAWTANQQLS peptide, which has been identified as presenting antioxidant activity [60]. Hydrolyzed chicken skin, from which four inhibitor peptides of the angiotensin-converting enzyme can be extracted, has also been identified as a source [51]. Blood from bovine and porcine sources has also been evaluated; it has been reported that hemoglobin and plasma hydrolysates exhibit antihypertensive, antioxidant, antimicrobial and opioid activity [61]. Table 1 shows some bioactive peptides obtained from food by-products and their corresponding activity.

For the choice of a suitable protein source, the use that hydrolysate is going to have must be considered, as is the added value of the final product with respect to the initial substrate. For example, to obtain hydrolysates with gelling and emulsifying properties, collagen and gelatin are often used due to their ability to form transparent gels [55,62], but proteins from eggs, meat, blood, viscera, and even cereals have also been used. As a source of fermentation for the growth of microorganisms, yeast or casein hydrolysates have been used. They are also sources for hydrolysates that are used in cosmetics. When a hydrolysate is intended to be used as a nitrogen source, fish proteins and microbial proteins are used in animal feed and soy and dairy proteins (especially whey proteins, the ideal raw material for the preparation of infant foods and enteral diets) are used in human nutrition [63,64,65].

## 3. Production of Peptides

Bioactive peptides can be found in most food products rich in proteins, and the consumption of foods with higher contents of proteins increases the possibility of acquiring them; nonetheless, their release is achieved through various mechanisms: (1) heat, (2) acid–base conditions in which the protein is hydrolyzed, (3) enzymatic hydrolysis, and (4) microbial activity (mainly in fermented foods) [66]. Once the structure of the bioactive peptides is known, it is also possible to synthesize them via chemical synthesis, recombinant DNA technology, and enzymatic synthesis [67]. Figure 1 shows the general process used for the production and identification of antioxidant peptides.

### 3.1. Enzymatic Hydrolysis

The enzymatic hydrolysis of proteins is the most common way to produce bioactive peptides. Once the protein source is selected, the hydrolysis is carried out using one or multiple specific proteases for the release of the peptides of interest. The factors that influence the release of the peptides are the duration of hydrolysis, the degree of protein hydrolysis, the enzyme–substrate ratio, and the treatment of the proteins (prior to hydrolysis) [71]. A great number of known bioactive peptides have been produced using a digestive enzymatic process and different combinations of proteinases such as pepsin, trypsin, alcalase, chymotrypsin, pancreatin and thermolysin.

Some studies have shown that biologically active peptides can be produced via the hydrolysis of milk proteins by digestive enzymes [72,73]. Pepsin, trypsin, and chymotrypsin are important enzymes that have been demonstrated to release antihypertensive peptides, phosphopeptides bonded to calcium, antibacterial peptides, immunomodulators, opioid peptides, and serum proteins (α-lactalbumin, ß-lactoglobulin and glycomacropeptide [19,73,74,75,76]. Likewise, angiotensin-converting enzyme inhibitor peptides (and algae hydrolysates, which have also been reported as a source of bioactive peptides [21]) have been obtained from the enzymatic digestion of proteins from marine products such as tuna, sardine muscle, and mackerel [28,36,77,78]. Alcalase, thermolysin and subtilysin are examples of other proteolytic enzymes that have been employed to release various bioactive peptides [73,79], such as inhibitors of the angiotensin-converting enzyme [73,80,81,82,83], antibacterial agents [84,85], antioxidants [86], immunomodulators [87] and opioids [73,88].

The advantages of enzymatic hydrolysis are that it is more easily reproducible because it allows for full process control [89], it is easy to scale up and it generally has a shorter reaction time than microbial fermentation [90], it can be optimized through the control of physical–chemical parameters such as pH or temperature to enable the ideal conditions for proteases to act [91], and it can be carried out by proteases of animal, vegetable, or food-grade microbials. Additionally, other proteases, such as microbial proteases that have different specificities for substrates and can release different bioactive peptides, can be sequentially used. An example is the Alcalase^®^ enzyme from *Bacillus licheniformis*, which has been used to obtain peptides with a good antioxidant, hypocholesterolemic and hypoglycemic profiles [92,93]. The proteases also have advantages in the generation of well-defined peptide profiles that have a high productivity rate and do not generate molecules that can harm health, as happens with chemical hydrolysis, which makes them viable for applications in the formulation of functional foods or nutraceuticals [94].

However, it is necessary to optimize the temperature and pH for each of the proteases [90]. Furthermore, both the choice of the protease used and the enzymatic hydrolysis time are important in deciding the type of peptides generated. Likewise, the stability of bioactive peptides produced during the phases of absorption, distribution, and metabolism must be evaluated to guarantee their stability in vivo [95]. To produce bioactive peptides, an international consensus methodology that simulates the physiological conditions of human gastrointestinal digestion in its different phases (oral, stomach and intestinal) has been established to guarantee the stability of bioactive peptides and facilitate their arrival to the systemic pathway where they can exert their therapeutic effects [96].

### 3.2. Microbial Fermentation

In this process, starter cultures are employed for the inoculation of microbial strains within a reactor that contains homogenous mixtures of concentrated proteins, water, and microbial nutrients such as simple sugars, which can be used for the growth and development of the strains [97] The protein breaks mainly due to the activity of the microbes and the peptidases secreted by them during fermentation [98]. Lactic acid bacteria have been mainly used for this purpose due to their high proteolytic activity. Some lactic acid bacteria employed as starter and non-starter cultures belong to strains of *Lactococcus lactis*, *Lactobacillus helveticus* and *Lactobacillus delbrueckii* ssp. *bulgaricus*; in their characterization processes, the participation of proteinases, coupled to the bacterial cell wall, has been observed, as has the presence of intracellular peptidases, including endopeptidases, aminopeptidases, tripeptidases and dipeptidases [99]. The type, quantity and activity of the peptides produced depend on the specific cultures employed. Peptides obtained via lactic acid bacteria from fermented goat milk have been reported to reduce oxidative stress and teratogenicity in human beings [100]. It has been reported that lactic acid bacteria used in yogurt fermentation produce peptides that can stabilize reactive oxygen species and inhibit lipoperoxidation [101]. Peptides from *Leuconostoc mesenteroides* ssp.; strains of *Cremoris*, *Jensenii lactobacilli* (ATCC25258), and *Lactobacillus acidophilus* (ATCC4356); and strains isolated from a whey fermentation system have also exhibited the inhibition of free radicals and lipoperoxidation [102]. In the same way, the heptapeptide HFGDPFH has been obtained from fermented mussel meats and exhibited high free radical scavenging activity [103].

Additionally, co-cultures with different combinations of bacteria, yeasts and fungi can be used to modulate hydrolysis processes, but the degree of hydrolysis depends on the fermentation type, time, microbial strain, and the protein source [104]; nevertheless, soybean proteins fermented by *Bacillus subtilis* MTCC5480 have been shown to produce a higher degree of hydrolysis than *B. subtilis* MTCC1747 [105].

Furthermore, this process is strongly influenced by environmental conditions such as humidity, pH, water oxide index, temperature, the availability of the substrate, the availability of free oxygen, and the size of the particles. To study the process on a pilot scale, researchers have developed bioreactors where heat and mass transfer are monitored, but this option is extremely expensive for commercial use. Likewise, the high density of the substrate makes it difficult to separate the biomass and therefore increases the energy required by the enzymes to produce metabolites [106].

### 3.3. Emerging Technologies for the Development of Antioxidant Peptides

Biochemical methods are the most common strategies to produce bioactive peptides; however, at an industrial level, a higher yield and lower cost is required; therefore, methods such as hydrolysis and technologies such as hydrostatic high-pressure, microwaves, and pulsated electric fields have been combined [107,108,109].

High pressure is a non-thermal treatment that applies pressures between 100 and 1000 MPa over a protein solution to modify the conformation and extension of the molecular chain of the proteins, aid the proteolysis reaction, and allow the enzymes to cleave peptide bonds in the new sites and produces bioactive peptides [110]. It has been observed that the high-pressure treatment improved the degree of hydrolysis, reducing power, and DPPH radical scavenging capacity of the peanut protein hydrolysate, with the effect of sequence on hydrolysis of 300 > 100 > 500 MPa (when all the treatments were 20 min) revealing that those subjected to 100 and 300 MPa were significantly higher (*p* < 0.05) than those treated with 500 MPa [109]. Likewise, it has been reported that the enzymatic hydrolysis of protein casein, under 100 MPa of pressure, improved the degree of hydrolysis and antioxidant properties compared to 200 MPa and atmospheric pressure with similar enzymatic hydrolysis [111]. These studies show that the treatment of proteins with high pressure improves the enzymatic hydrolysis and yield of antioxidant peptides, as well as bioactivity. Processing assisted by microwaves utilizes electromagnetic radiation, in a frequency range of 300 MHz–300 GHz, to heat a solvent in a sample more rapidly than conventional heating [57] due to inter- and intramolecular friction in conjunction with the movement and collision of ions, which provoke the rapid heating of the reaction and the rupture of the walls and cell membranes [112]. Microwave heating has been applied as a pretreatment followed by enzymatic hydrolysis, and it has been found that the use of this method increases the degree of hydrolysis, protein solubility, and radical scavenging, thus improving the accessibility and susceptibility of bonds to enzymes [110,113,114]. Studies have shown that peptides with antimicrobial activity have been obtained with pepsin from different variants of casein and from whey proteins [76]. Processing assisted by ultrasound uses the acoustic cavitation induced by ultrasound, which creates mechanical and thermal effects and modifies the physicochemical properties of proteins [115,116]. Some studies have provided evidence on the use of ultrasonic processing as a pre-treatment for the improvement of the protein hydrolysis procedure and to increase the antioxidant activity of the peptide hydrolysates. Wen et al. (2020) [108] employed type-S ultrasound microfrequencies as a pretreatment for the arrowhead protein for its posterior hydrolysis, and they showed that the hydrolysates obtained with double-frequency ultrasound (20/40 kHz) had an effect over the antioxidant activity of the hydrolysates without ultrasonic treatment. There was a 63.61% inhibition of DPPH• and a 65.11% inhibition of ABTS•+. The authors of [117] employed a 750 W ultrasound as pre-treatment, thermal treatment at 50 °C, and hydrolysis with alcalase for the procurement of collagen from salmon scales, and they reported a higher degree of hydrolysis and the higher inhibition of ABTS, DPPH, and ferric-reducing antioxidant power in comparison to other pre-treatments.

Another treatment consists of a pulsated electric field, consisting of a 10–50 kV/cm electric field that is executed as multiple short pulses (typically 1–5 μs) in frequencies of 0.2–0.4 MHz. The treated proteins expose hydrolysis sites that were previously inaccessible to peptidases [118], and the processing—carried out with subcritical water (it is an ecological technology)—employs the generated pressure to heat the water at a temperature between its atmospheric boiling point (100 °C and 0.1 MPa) and critical point (374 °C and 22.1 MPa); the yield and bioactivity of the hydrolysate of the treated proteins depend on the hydrolysis temperature and pressure [119]

Some authors such as Chian et al. (2019), Alvarez et al. (2016), Mikhaylin et al. (2017), and Ahmed et al. (2018) [118,120,121,122] have applied pre-treatments to obtain peptides, and these treatments have led to improved antioxidant and functional properties. For example, the use of ultrasound as a pre-treatment [123] and its simultaneous application during enzymatic hydrolysis [124] could modify protein structure and conformation, affecting the hydrogen bonds or hydrophilic interactions and the tertiary and quaternary structure thereof. In addition to the physical and mechanical effects, ultrasound can expose hydrolysis sites to a greater extent, resulting in an increase in susceptibility to the action of proteinases and causing an increase in the degree of hydrolysis. The adverse effects that may occur include damage to the conformation of the enzyme itself and therefore to its hydrolytic activity [115]. The advantages of microwave-assisted peptide synthesis over other conventional heating processes include its selectivity, allowing for digestion to be carried out in a short period of time [125,126] due to the selective heating of the intermolecular friction product of the alignment of ions and dipoles to the oscillating electric field of the microwaves and a subsequent dissipation of the energy via heating in the nucleus. This is also a clean process that leads to energy savings of between 25 and 75%. It allows for the control of parameters such as pressure, time and working temperature, and enables the obtainment of optimal results in shorter times. Experimental results have indicated that digestion temperature, reaction time, enzyme-to-substrate ratio, and digestion buffer are the primary factors required to optimize the method for successful hydrolysis [127].

Supercritical fluid extraction is a very useful technique as a method for the separation and pre-concentration of bioactive peptides. One advantage of using supercritical fluids is that they can penetrate a porous solid more easily, resulting in faster extraction. The dissolution of the supercritical fluid is controlled as a function of pressure and temperature. In supercritical extraction, fresh fluid is continuously passed through a sample. Supercritical fluids are easily recovered from the extract by depressurization. Supercritical extraction uses environmentally friendly solvents, and the supercritical fluid can be recycled or reused, minimizing the generation of waste.

However, disadvantages include that a supercritical fluid must have a series of properties in order to be used as a solvent in industry and especially in the food industry, e.g., high solvent capacity; selective, zero or low flammability; easy obtainment in high purity; low price; low or null toxicity; environment friendliness; non-corrosiveness; moderate critical conditions; and gaseous under ambient temperature and pressure conditions. In addition to obtaining food ingredients, ethanol, water and CO_2_, which is cheap and commercially available (even with high purity and declared Generally Recognized as Safe (GRAS) designation), must be used as modifiers. Due to the nature of the supercritical fluid extraction process, high pressures are required to carry out extraction. The method has a high cost due to investment for equipment and maintenance. Solvent compression requires elaborate recirculation measures to reduce energy costs. The use of modifiers can alter the polarity of the CO_2_, but they can remain in the extract, thus requiring a subsequent separation process [128,129]. Regarding concern for the environment, supercritical fluid extraction has shown excellent opportunities to achieve the key objectives of “green chemistry”, a term that refers to the design of chemical products and processes that reduce or eliminate the use or production of environmentally hazardous substances. This methodology meets several objectives such as the reductions in the use of solvents and lowering their potential harm as much as possible. The coupling of the aforementioned emerging technologies, together with enzymatic hydrolysis, have therefore proven to be a feasible method to produce bioactive peptides since they can be obtained in less time and at lower costs compared to the usual methods.

The effect of two pretreatments on the antioxidant activity in the antioxidant protein hydrolysates of quinoa has been evaluated using extraction with supercritical CO_2_ and ethanol as co-solvent; this type of pretreatment was compared to a conventional method of extraction with the ether of quinoa oil without the recovery of bioactive compounds [130]. Significant effects were found in the degree of hydrolysis (23.93%) and antioxidant activity (1181.64 μmol TE/g of protein) compared to the conventional method (24.33% and 1448.84 μmol TE/g protein, respectively). An economic evaluation was carried out using the SuperPro Designer 9.0^®^ simulator to estimate the manufacturing cost (MC) considering laboratory scale lots (1.5 kg/lot) and industrial scale (2500 kg/lot). A total investment in the range of 400,000–500,000 dollars was calculated for the industrial scale for the conventional extraction method and 900,000–1,000,000 dollars when using the supercritical fluid extraction method. This finding is very interesting because it provides a global overview of these processes, with which it will be possible to propose changes to make the processes more efficient and economical.

### 3.4. Bioinformatic Tools in Obtaining Bioactive Peptides

For the identification and characterization of bioactive peptides, researchers have used bioinformatics techniques, e.g., computer-based (in silico) simulations have been applied to discover encrypted bioactive peptides in food proteins [17,131,132,133].

The in silico approach uses information accumulated in databases (Table 2) in an analysis prior to the experimental study as a rapid prediction tool to discover structure–function relationships, predict peptide sequences likely to exhibit specific activities, locate encrypted peptides in protein sources, anticipate the release of these fragments by the action of specific enzymes, and simulate or optimize processes of protein hydrolysis in order to predict the amino acids or peptides released when hydrolyzing the sequence of a protein with one or more enzymes and to identify peptides with biological activity and predict bioactive properties of peptides [17,134,135]. This approach also allows for the simultaneous evaluation of multiple food proteins and proteolytic enzymes to determine the frequency of occurrence of cryptic bioactive peptides in the primary structure of food proteins.

Protein sequences can be obtained from databases. In particular, the SwissProt and TrEMBL databases of the universal protein knowledge base (UniProtKB) (Table 2) can be used to calculate the release frequency of bioactive peptide fragments via a/N, where “a” is the number of bioactive peptides with specific activity released by the action of a certain enzyme and “N” is the total number of amino acid residues in the protein, and to simulate the proteolytic specificities of enzymes for generate in silico peptide profiles because in silico proteolysis such as BIOPEP, ExPASy PeptideCutter (http://web.expasy.org/peptide_cutter, accessed on 28 January 2022) and PoPS (http://pops.csse.monash.edu.au, accessed on 28 January 2022) programs include “enzyme action” tools. These tools include databases of peptide sequences with various bioactivities and programs for the simulation and prediction of peptide bioactivities [135].

Table 2 shows several peptide databases, some of which are dedicated to food peptides. These bases provide the collection of peptides with diverse bioactivities and sequences with different biofunctions.

The peptides resulting from in silico proteolysis can then be matched with bioactive peptides in databases for predetermined bioactivities. This approach has recently been used to study the distribution of antimicrobial peptides, antioxidants, inhibitors of disease-related enzymes within the primary structure of typical food proteins [132,158], and precursors of bioactive peptides [133], as well as to obtain sustainable proteins.

Barrero et al. (2021) [159] carried out a quantitative comparison of the bioactive peptides obtained from the caseins and seroproteins present in the milk of bovines (*Bos taurus*), sheep (*Ovis aries*), goats (*Capra hircus*) and buffalo (*Bubalus bubalis*) from in silico digestion processes catalyzed by proteases present in the human digestive system: pepsin (EC 3.4.23.1), trypsin (EC 3.4.21.4) and chymotrypsin (EC 3.4.21.1). The characterization of bioactive peptides and in silico digestion was performed using BIOPEP-UMW and quantitative evaluation from the calculation of release frequencies. The results showed 9 stimulant peptides for the four species and a slightly higher mean frequency for the buffalo with a value of 0.096, 7 renin inhibitor peptides with a mean frequency of 0.0074 for cattle and buffalo, 68 DPP4 inhibitor peptides with a frequency mean of 0.0716 for goats and sheep, 10 antioxidant peptides with a mean frequency of 0.0105 for cattle, 41 ACE inhibitor peptides with a mean frequency of 0.0432 for goats and sheep, and 8 DPP3 inhibitory peptides for the cattle, goats, and buffalo—the last with a slightly higher average frequency with a value of 0.0085. One peptide with hypocholesterolemic action and one peptide inhibitor of CaMPDE were obtained from the 𝛽-lactoglobulin of the four species.

Panjaitan et al. (2018) [160] attempted to match sequences identified in a tryptic hydrolysate of giant grouper (*Epinephelus lancetolatus*) roe proteins with bioactive peptide sequences listed in the BIOPEP-UWM database. Using this research strategy, it was reported that peptides found in giant grouper roe protein hydrolysates mainly showed ACE inhibitors, DPP-IV, and antioxidant bioactivities. Once in silico studies have been carried out, it is expected that the best combinations of protein proteases will be followed up and selected in order to experimentally validate the production and bioactivity of the peptides. In this study, the results of BIOPEP-UWM’s in silico and in vitro hydrolysis showed similar sequences to vitellogenin.

In silico studies applied to peptides can reduce the time required to detect bioactive peptides present in various protein sources using various proteases and can lead to the discovery of new and sustainable precursors of known bioactive peptides. However, it is not guaranteed that peptides in silico can be experimentally reproduced due to the complex nature of protease–protein interactions. Another disadvantage is that the currently used bioinformatic approach does not consider other peptide sequences within “in silico hydrolysates” that do not currently exist in bioactive peptide databases. Furthermore, the information that is available in databases often involves well-characterized and-purified proteolytic enzymes compared to enzymes used commercially for food processes that are less substrate-specific and of variable purity [161]. Furthermore, currently existing databases are often not regularly updated and can become unreliable, especially with information on new bioactive peptides constantly accumulating.

With the vast amount of information on the preclinical bioactivities of food-derived peptides, future efforts should consider exploring opportunities to promote the use of these peptides as functional ingredients to develop products with targeted health benefits, which could emphasize improving the taste, gastric stability, and bioavailability of peptides; providing clinical evidence to support their health effects; optimizing their inclusion in food products to limit undesirable reactions; and sourcing sustainable protein raw materials for their production in an attempt to reduce the heavy dependence on primary human food.

## 4. Health Benefits

Some studies have shown that bioactive peptides have beneficial effects on health, although the relationship between chemical structure and biological activity has not been predicted to date since the activity of a peptide depends on its structure, the composition of its amino acids, the type of terminal amino acid, the length of the peptide chain, the charge of the amino acids that make up the peptide, and the hydrophobic and hydrophilic characteristics of the amino acid chain, among others [21,133].

### 4.1. Antioxidant Activity

The biological activity of peptides has been widely studied, since, in addition to their antioxidant activity, they have nutritional and functional characteristics.

Antioxidant peptides are purified and linked amino acids from hydrolysates. They contain 5–16 amino acid residues, and those that come from food are considered safer because they are healthy compounds with low molecular weight, low cost, high activity, and easy absorption. They have advantages over enzymatic antioxidants because they have a simpler structure, greater stability, and do not generate dangerous immunological reactions [162,163,164].

Antioxidant peptides can be used as hydrolysates of the precursor proteins or as antioxidant peptides. A hydrolysate is a mixture mainly composed of peptides and amino acids that are produced through the hydrolysis of proteins by treatment with enzymes, acids, alkalis or fermentation. There is a classification by degree of hydrolysis, and this determines applications. For instance, hydrolysates with a low degree of hydrolysis and better functional characteristics [165] and hydrolysates with various degrees of hydrolysis can be used either as emulsifiers (such as in mayonnaise, minced meat, sausage, or ice cream) [166,167,168], as flavorings and protein supplements or in medical diets [169,170]. Furthermore, hydrolysates with a high degree of hydrolysis are used primarily as nutritional supplements and in special medical diets.

The 20 amino acids present in proteins can react with free radicals; the most reactive are those that include the sulfur amino acids M and C; the aromatics W, Y, and F; and an imidazole ring such as H. According to Girgih et al. (2015) [171], the last group comprises amino acids that have a high antioxidant capacities due to the imidazole rings that they use as a proton donors. Histidine and the indole ring of tryptophan, as well as lysine and valine, can scavenge hydroxyl radicals by acting as electron donors that generate a hydrophobic microenvironment in the molecule, which favors the antioxidant capacity of the peptide [27,123].

Various studies have been carried out to investigate the antioxidant properties of hydrolysates or bioactive peptides from plants and animals. Table 3 presents a list of various peptide fractions with antioxidant activity.

#### Mechanisms of Antioxidant Peptides

Several studies have shown that antioxidant peptides can act as (1) chelators of a metal, (2) radical inhibitors, or (3) physical shielding [103,191,192,193]. In Figure 2, these mechanisms are shown.

Mechanism 1, as chelators of a metal: These peptides inhibit the production of free radicals by stabilizing metallic pro-oxidants through sequestering agents. The carboxyl and amino groups of the side chains have a chelating function of metal ions due to their ability to dissociate and be proton donors. It has also been reported that chelating peptides are rich in histidine and prevent oxidative activity by chelating metal ions. Copper-chelating peptides are rich in histidine and prevent copper’s oxidative activity by chelating metal ions. The imidazole ring of this residue is directly involved in binding to copper. These peptides have also been found to be rich in arginine. Although this amino acid lacks chelating properties, it may favor the binding of the peptide to metal ions. Therefore, copper chelating peptides may be useful by preventing not only copper oxidative activity that can damage cells in the luminal space of the stomach but also the copper-induced oxidation of LDL if it reaches the bloodstream. They can also be useful in organs such as the brain, where the oxidative process is involved in the development of certain diseases [103,194,195]. R, a basic amino acid, confers peptide charge and hydrogen-bonding interactions, which are essential properties for combination of with the abundant anionic component of the bacterial membrane, and it appears that W residues play the role of natural aromatic activators of R-rich antimicrobial peptides via π-ion pair interactions, thus promoting enhanced peptide–membrane interactions [196].

Mechanism 2, as radical inhibitors: Peptides stabilize radicals by donating electrons, maintaining their own stability via the resonance of their structure. Some reported peptides are those that contain aromatic amino acids such as Y, H, W, and F and tripeptides with C that contain SH groups and are radical scavengers. In an oxidizing environment where unsaturated lipids are present, active peptides are more inclined to donate electrons than lipids and form more stable radicals or non-reactive polymers through radical–radical condensation [194,197,198].

Mechanism 3, physical shielding: Peptides inhibit lipid peroxidation by acting as a physical barrier or membrane. They are surfactant components and can split at the oil–water interface, forming a thick membrane or coating to avoid the direct contact of lipids and radicals with other oxidizing compounds [199].

The amino acid sequence and the presence of certain amino acid residues, such as histidine, tyrosine, tryptophan, methionine, cysteine, and proline, have also been shown to be determining factors in the antioxidant capacity of peptides [199]. The relationship between the presence of certain amino acids and their respective antioxidant activity is shown in Table 4.

### 4.2. Antimicrobial Activity

Antimicrobial peptides (AMPs) are molecules comprising 10–55 amino acids; they can be neutral, aniconic, or (mainly) cationic, and they have important innate antibacterial and antifungal properties. Almost 50% are hydrophobic and have a molecular weight of less than 10 kDa; they tend to be heat-stable due to the presence of fewer amino acids and can be generated in vitro via enzymatic hydrolysis [199,200,201,202,203].

It has been reported that the main mechanism of action of AMPs against microorganisms is the rupture of membranes, is which an electrostatic union with the membrane that carries negative charge is initiated through the positive charge, therefore leading to the leakage of metabolites, membrane depolarization, interruption of respiratory processes and finally, and cell death [204].

The mechanism of action of AMPs is divided into three stages:Stage 1. Attraction for the bacterial cell wall

Antimicrobial peptides are attracted to bacterial surfaces thanks to their amphipathic structure, the positive charge at physiological pH through electrostatic bonding between anionic or cationic peptides, and structures on the bacterial surface. It is likely that cationic AMPs are first attracted to the net negative charges that exist on the outer envelope of Gram-negative bacteria (e.g., anionic phospholipids and lipopolysaccharides (LPS)) and to teichoic acids on the surface of Gram-positive bacteria, acting as detergents that break down the bacterial membrane [205,206].Stage 2. Union to the cell membrane

Once on the microbial surface, peptides must pass through capsular polysaccharides before they can interact with the outer membrane, which contains LPS in Gram-negative bacteria and transverse capsular polysaccharides, teichoic acids, and lipoteichoic acids [207].Stage 3. Peptide insertion and cell membrane permeabilization

Peptides interact with the cytoplasmic membrane in lipid Gram-positive bacteria and then gain access to the cytoplasmic membrane and can interact with lipid bilayers [207]. This interaction allows for the accumulation of intramembrane peptides until reaching a threshold at which collapse occurs and the membrane ruptures.

The potential mechanisms of action of antimicrobial peptides include:Bacterial membrane permeabilization: The mixed hydrophobic and cationic composition of antimicrobial peptides makes them highly suitable for interacting with and disrupting microbial cytoplasmic membranes that present anionic, lipid-rich surfaces, such as phosphatidylglycerol and cardiolipin. The fact that all Gram-negative and Gram-positive bacteria show this type of negatively charged lipids explains the lack of specificity of most antimicrobial peptides, promotes the attraction between antimicrobial peptides and bacterial membranes, and prevents their binding to most host cell membranes. The selective toxicity of antimicrobial peptides is based on differences in the membrane potential of mammalian microbes and cells. Microbes tend to have a significantly large charge difference across their membranes compared to mammalian cells, favoring cationic defensins to selectively attack microbes [208,209,210,211].Enzymatic attack on bacterial wall structures: Several epithelial antimicrobial peptides kill bacteria through enzymatic attacks on key cell wall structures. Lysosomes are effective against Gram-positive bacteria, where peptidoglycan is more accessible, than against Gram-negative organisms, where peptidoglycan is protected by the outer membrane [207]. In addition to directly killing bacteria, the enzymatic activity of lysozymes can regulate the innate immune responses to certain microorganisms. Secretory phospholipase A2 (sPLA2) is an example of an antimicrobial peptide that kills bacteria through an enzymatic mechanism. Bacterial membranes, rich in phosphatidylglycerol and phosphatidylethanolamine, are the key targets of sPLA2, but the enzyme can break down other phosphotriglyceride substrates. The sPLA2 peptide penetrates the bacterial cell wall to access the membrane, where it hydrolyzes phospholipids and thus compromises the integrity of the bacterial membrane. This enzyme is bactericidal, with preferential activity against Gram-positive bacteria [212].Interference at the intracellular level: It seems likely that many AMPs can translocate across microbial membranes at concentrations that do not induce permeabilization. Once in the cytoplasm, they can attack DNA and chaperonins, alter the formation of the cytoplasmic membrane septum, inhibit cell wall synthesis, reduce nucleic acid synthesis, suppress protein synthesis, or inhibit enzyme activity [213,214].Antimicrobial peptides that exploit multiple antimicrobial strategies: The peptide can affect microbes in ways other than cell destruction such as the destabilization of the membrane, the filamentation of bacterial cells due to the insertion of peptides in the membrane, the blocking of DNA replication, and the inhibition of membrane proteins involved in septum formation [214], lectin-like behavior, and antitoxin activity.

In addition to their antimicrobial capacity, peptides are also effective against fungal infections since they suppress the reproduction or growth of fungi. Antifungal peptides are differentiated by their mechanisms of action: the first group are amphipathic peptides—lytic peptides that break down membranes, are naturally abundant, and have hydrophilic and hydrophobic elements—that have two surfaces; one is positively charged and the other uncharged (neutral). Some penetrate cells and can alter the structure of a membrane without crossing the membrane simply by attaching to its surface. The second group of peptides can obstruct the synthesis of the cell wall or the biosynthesis of glucan or chitin, essential cellular components [215,216].

In medicinal applications, antimicrobial peptides are preferred over conventional bactericidal antibiotics because they kill bacteria faster and are not affected by antibiotic-resistance mechanisms [217,218,219]. Various studies have reported the presence of antimicrobial peptides in milk and breast secretions, as well as from other food sources such as derivatives of rice protein, soy protein, buckwheat, and pepper [87,220,221]. Additionally, peptides derived from casein have shown antimicrobial activity in in vitro models in the presence of various microorganisms such as *Staphylococcus* ssp., *Streptococcus pyogenes*, *Diplococcus pneumoniae* and *Bacilus subtilis*. The presence of casecidin, obtained from α-s1 casein, after being hydrolyzed by chymosin, inhibits the growth of these microorganisms at high concentrations [84].

Recent research [222,223] has shown that are peptides that exist in different organisms such as amphibians and marine animals could be a source of peptides with antimicrobial activity, e.g., the amphibian *Rana cancrivora* is the main source of carcinine [224]. On the other hand, some marine organisms present peptides with both antimicrobial and antibacterial activity, among which are AMPs that play a key role in the strategy against infections [225]. *Bacillus subtilis*-fermented fish meat protein hydrolysates prepared from sardinella, zebra blenny, goby, and ray fish showed antibacterial activity, and sardinella hydrolysates were the most effective against Gram-positive bacteria [226]. AMPs are small oligopeptides generally containing 10–100 amino acids, with a net positive charge and an amphipathic structure; they are the most important components of the antibacterial defense of organisms at almost all levels of life: bacteria, fungi, plants, amphibians, insects, and mammals. Examples from various sources are shown in Table 5.

## 5. Bioavailability of Bioactive Peptides

A protein that is ingested through the diet is digested and absorbed in the gastrointestinal tract; it is denatured by gastric acid and hydrolyzed into small peptides and amino acids by gastric and pancreatic proteases. Some amino acids are used in the cell by the enterocytes themselves as a source of energy, and others undergo metabolic transformations before passing into the blood during cell turnover (Figure 3) [243]. The final step in the digestion of dietary proteins occurs on the surface of enterocytes via the action of intestinal brush border cell proteases [244]. Once the amino acids and small peptides that have managed to cross the intestinal barrier, enter the blood, and reach the liver through the portal circulation, they are captured and used by the liver; then, some enter the systemic circulation and are used by the tissue’s 10 peripherals. The metabolic life of amino acids is complex and starts with their use as energy or gluconeogenic substrates and ends with the synthesis of new proteins and peptides [243].

The potential beneficial effects of biopeptides depend on their ability to reach the organs where they are going to perform their function intact. Therefore, it is important to consider the in vivo differences of bioactive peptides, since peptides must be able to overcome barriers and actively reach their target because they are subject to degradation and modification in the intestine, vascular system, and hepatic system. Although peptides are rapidly metabolized to their constituent amino acids, some studies have shown that several peptides are resistant to these physiological processes and can reach the circulation [246].

Gardner (1998) [247] indicated that a small portion of bioactive peptides can pass the gut barrier, and, though too small to be considered nutritionally important, this proportion can have biological effects at the tissue level. However, peptides and proteins can escape digestion and be absorbed intact. One reported mechanism for intact peptide uptake is the use of paracellular water-soluble peptides. Hydrophobic peptides are absorbed by diffusion through tight junctions between cells by a passive diffusion process independent of energy [248]. Some small peptides, resistant to hydrolysis, leave the enterocyte and enter the circulatory system through a peptide transporter located in the intestinal basolateral membrane; in endocytosis [248], large polar peptides bind to cells and are absorbed into the cell through vesiculation and the lymphatic system [249]. Highly lipophilic peptides, too large to be absorbed into circulation, are absorbed from the interstitial space into the intestinal lymphatic system.

Molecular size, structural properties, and hydrophobicity also affect the major peptide transport pathway [250]. Research indicates that peptides with 2–6 amino acids are more easily absorbed than proteins and free amino acids [251].

Roberts et al. (1999) [252] reported that small (di- and tripeptides) and large (10–51 amino acids) peptide chains can cross the intestinal barrier intact and exhibit their biological functions at the tissue level. However, as the molecular weight of peptides increases, their ability to pass through the intestinal barrier decreases.

Therefore, it can be deduced that due to incomplete bioavailability of a peptide after oral ingestion, a peptide with pronounced bioactivity in vitro may exert little or no activity in vivo. However, other routes of derivation can increase the possibility of absorption of peptides, diminishing the problem [56].

Vilcacundo et al. (2017) [253] extracted a protein concentrate from quinoa that was digested under in vitro gastrointestinal conditions. Quinoa proteins were almost completely hydrolyzed by pepsin at pH 1.2, 2.0 and 3.2. At high pH, only partial hydrolysis was observed. During the duodenal phase, no intact proteins were seen, indicating their susceptibility to simulated digestive conditions in vitro. The zebrafish larval model was used to assess the in vivo the ability of gastrointestinal digests to inhibit lipid peroxidation. Gastric digestion at pH 1.2 showed the highest percentage of the inhibition of lipid peroxidation (75.15%). Lipid peroxidation increased after the duodenal phase. These peptides could be further hydrolyzed by pancreatin, releasing new fragments with greater antioxidant activity. Thus, the hydrolysate obtained at the end of the digestive process showed a higher inhibitory capacity with an inhibition percentage of 82.10%, similar to that of the positive control butylated hydroxytoluene used in the test (87.13%). The results of this study demonstrated, for the first time, that quinoa protein hydrolysates obtained during in vitro simulated gastrointestinal digestion were capable of inhibiting lipid peroxidation in the in vivo zebrafish model.

Carrillo et al. (2016) [254] identified five peptides (VAWRNRCKGTD, IRGCRL, WIRGCRL, AWIRGCRL, and WRNRCKGTD) from hen egg-white lysozyme with in vitro antioxidant activity (1970, 3123, 2743, 2393 and 0.013 µmol Trolox/µmol peptide, respectively), being the last one with less activity. They determined the inhibition of lipid peroxidation in an in vivo model of zebrafish larvae through cell damage. This assay confirmed that antioxidant peptides of hen egg-white lysozyme were not toxic for the zebrafish larvae, which were normal after 24 h of assay. When the zebrafish larvae were examined, no morphological abnormalities, such as crooked bodies, spinal deformities, or any significant effects in the growth of the body, were observed. The lysozyme peptides were shown to efficiently inhibit the lipid peroxidation in zebrafish larvae. The AWIRGCRL peptide demonstrated 63.2% thiobarbituric acid reactive substance (TBARS) inhibition. A higher antioxidant activity was shown the of WIRGCRL, AWIRGCRL and WRNRCKGTD peptides. In the case of the TBARS inhibition, peptide size probably contribues to the increased values of activity in the IRGCRL, WIRGCRL, AWIRGCRL, and WRNRCKGTD peptides.

In vivo, the bioavailability of a bioactive peptide depends on its resistance to hydrolysis by peptidases present in both the intestinal tract and the serum of the bloodstream, as well as the ability of the generated peptides to be absorbed through barriers such as the intestinal epithelium. The in vitro method tends not to consider some aspects such as the absorption, distribution properties, metabolism, and excretion of peptides, which can result in low activity and bioavailability.

The biological effects of peptides depend on their bioavailability, which is determined by their resistance to gastrointestinal digestion [255]. However, to ensure the biological effects of peptides, it is important to carry out in vitro and in vivo studies that confirm their stability, absorption capacity, and mechanism of action, which increase and diversify their nutraceutical and pharmaceutical applications.

According to what is stated in the present review, bioactive peptides present in the diet, could improve the body’s ability to regulate oxidative stress, which cannot be modified by the intervention of endogenous antioxidant defenses and thereby help to reduce the incidence of some degenerative diseases.

## 6. Conclusions

The objective of this review was to provide an overview of foods and by-products as sources rich in bioactive peptides such as antioxidants and antimicrobials, as well as their potential in reducing oxidative stress through exogenous factors or the consumption of antibiotics for the improvement of the health of consumers.

The production of these peptides can occur through enzymatic processes or via fermentation. New extraction technologies with supercritical fluids, microwaves, and pulse electric field are being used as pretreatments combined with conventional methods to obtain higher yields and lower costs of these bioactive compounds, with significant effects on the degree of hydrolysis, antioxidant activity, and antimicrobial activity. One of the findings observed in this review was from economic study, in which it was observed that obtaining these compounds in combination with emerging technologies enabled reductions in costs at the industrial level compared to conventional methods. Furthermore, the peptides resulting from in silico proteolysis can then be matched with bioactive peptides in databases for predetermined bioactivities.

In silico studies of peptides have been able to reduce the time required to detect bioactive peptides present in various protein sources using various proteases and to lead to the discovery of new and sustainable precursors of known bioactive peptides.

With the vast amount of available information on the preclinical bioactivities of food-derived peptides, future studies should consider exploring opportunities to promote the use of these peptides as functional ingredients to develop products with targeted health benefits, which could emphasize improving the taste, gastric stability, and bioavailability of peptides; providing clinical evidence to support their health effects; optimizing their inclusion in food products to limit undesirable reactions; and sourcing sustainable protein raw materials for their production in an attempt to reduce the heavy dependence on primary human food. However, the constant update of these approaches is necessary.

Regarding in vivo and in vitro studies, peptides obtained from some cereals and enzymes derived from by-products have shown the ability to inhibit lipoperoxidation.

There have been numerous studies on the antimicrobial peptides that can be used as protectors against oxidative stress due to the consumption of antibiotics, and deeper in vivo studies on the exact mechanism of action are recommended. Additionally, antioxidant and antimicrobial peptides could be used to produce functional foods. Therefore, it is also important to study the interaction of peptides with other food constituents, as well as the effects of incorporating these peptides and the processing conditions on the bioactivity of peptides in food matrices.

## Figures and Tables

**Figure 1 molecules-27-01343-f001:**
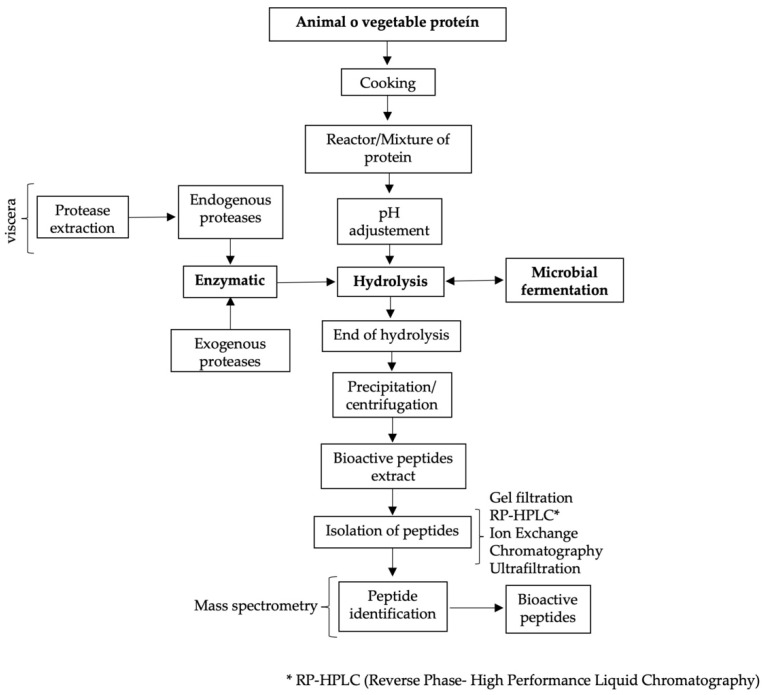
Diagram of the bioactive peptide production process [68,69,70].

**Figure 2 molecules-27-01343-f002:**
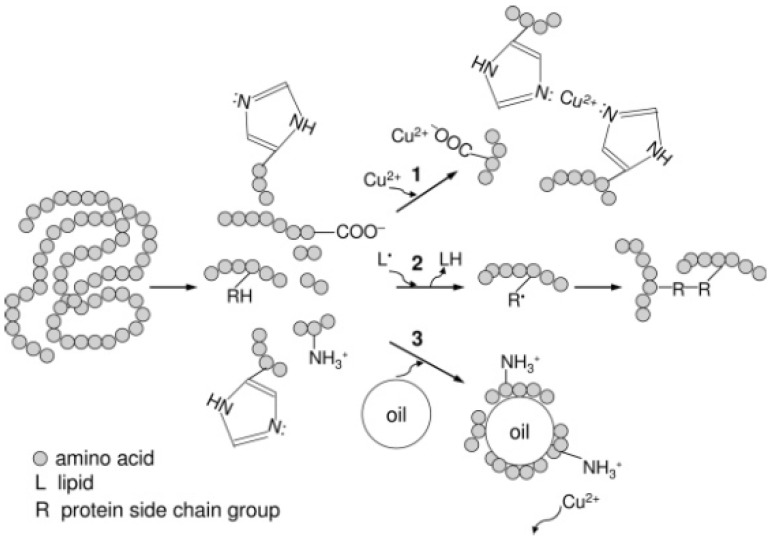
Schematic representation of the chemical and physical mechanisms of antioxidant peptides, used to inhibit oxidative processes: (1) chelators of metals, (2) radical inhibitors, and (3) physical shielding [194].

**Figure 3 molecules-27-01343-f003:**
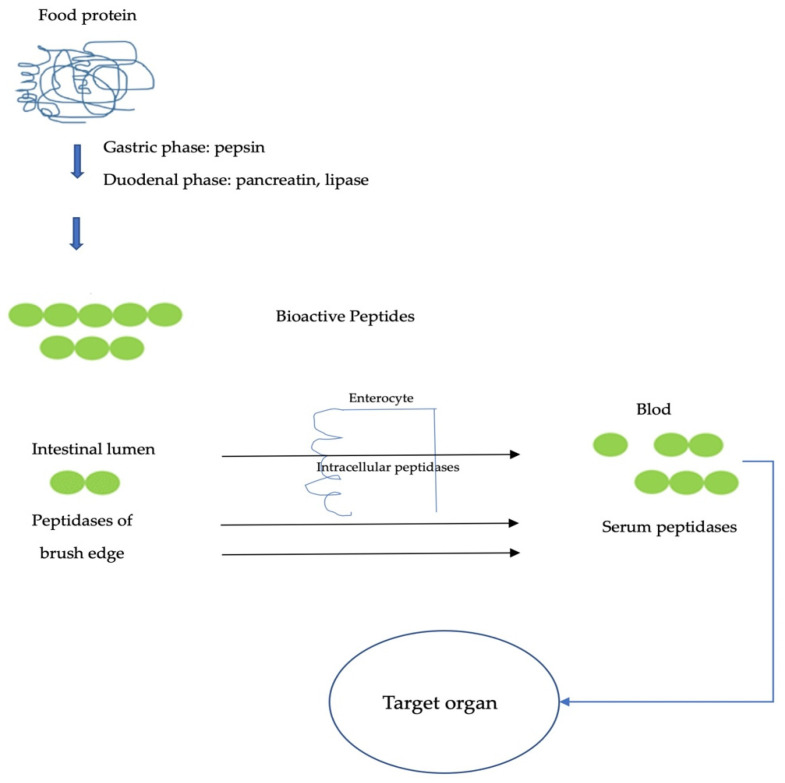
Diagram of protein digestion and transepithelial transport of peptides until they reach circulation [245].

**Table 1 molecules-27-01343-t001:** Bioactive peptide sources from various foods and their activity.

Activity	Source	Protein of Origin	Bioactive Peptide o Sequence
Inhibition of angiotensin-converting enzyme (ACE) and antihypertensive	Soy	Soy protein	NMGPLV
	Fish	Muscle protein	LKP, IKP (derived from sardines, mackerel, tuna, squid)
	Meat	Muscle protein	IKW, LKP
	Milk	α-LA, β-LG α-, β, ƙ-CN	Lactokinins (WLAHK, LRP, LKP)
	Egg	Ovotransferrin Ovalbumin	Ovokinin (FRADHPPL), Ovokinin (2–7) (KVREGTTY)
	Wheat	Gliadins	Casokinins (FFVAP, FALPQY, VVP)
	Broccoli	Encrypted protein	IAP
	Chicken skin	Collagen	YPK
	Chicken legs	Collagen	GAHpGLHpGP
Immunomodulator	Rice	Rice albumin	Orizatensinin (GYPMYPLR)
	Egg	Ovalbumin	Unspecified peptides
	Milk	α-β-ƙ-CN α-LA	Immunopeptides (αs 1 TTMPLW)
	Wheat	Gluten	Immunopeptides
Cytomodulator	Milk	α-β-CN	A-casomorphin (HIQKED (V)), β- casomorphin-7 (YPFPGP)
Opioid agonist	Milk	α-LA, β-LG α-, β-CN	α-lactorfin, β- lactorfin Casomorphin
	Wheat	Gluten	Gluten exorphins A4, A5 (GYYPT), B4, B5 and C (YPISL)
Opioid antagonist	Milk	Lactoferrin ƙ-CN	Lactoferricin Caxosines
Antimicrobial	Egg	Ovotransferrin	OTAP-92 (f109-200)
	Milk	Lysozyme α-β-ƙ-CN	Unspecified peptides
	Bovine cruor	Lactoferrin	Lactoferricin Caxosines TSKYR
	Guava seeds	Glycine	Pg-AMP1
	*Vicia faba* seeds	Hydrolysate seed proteins	LSPGDVLVIPAGYPVAIK, EEYDEEKEQGEEEIR
	Tomato pomace	Hydrolysate proteins	Unspecified peptides
	Rice bran	Rice bran proteins	KVDHFPL
	Bovine hemoglobine	Hemoglobine	(F)VNFKLLSHSLL, (L)TSKYR, (F)KLLSHSL, (L)QADFQKVVAGVANALAHRYH, MLTAEEKAAVTAFWGKVKVDEVGGEALGRL
Antithrombotic	Milk	ƙ-CN (glycomacropeptide)	ƙ-CN (f106-116) a, casoplatelin
Mineral-carrying Anticarcinogenic	Milk	ƙ-CN (glycomacropeptide)	ƙ-CN (f106-116) a, casoplatelin
Hypocholesterolemic	Soy	Glycinin/conglycinin	LPYPR
	Milk	β-LG	IIAEK
Antioxidant	Sardine	Sardine muscle	MY
	Wheat	Wheat germ protein	Unspecified peptides
	Milk	α-LA, β-LG	MHIRL, YVEEL, WYSLAMAASDI
	Tuna	Tuna bones	VKAGFAWTANQQLS
	Oyster	Oyster by-products	PVMGD
	Leatherjacket	Leatherjacket heads	EHGV
	Salmon	Salmon by-products	WEGPK, GPP, GVPLT
	Soy	Glycinin /conglycinin	LLPHH, VNHDHQN, LVNHDHQN, LLPHH
	Rice endosperm	Rice protein	FRDEHKK, KHDRGDEF

CN: casein; LA: lactoalbumin; LG: lactoglobulin; f: fragment; A: alanine; R: arginine; N: asparagine; D: aspartic acid; C: cysteine; E: glutamic acid; Q: glutamine; G: glycine; H: histidine; Hp: hydroxyproline; I: isoleucine; L: leucine; K: lysine; M: methionine; F: phenylalanine; P: proline; S: serine; T: threonine; W: tryptophan; Y: tyrosine; V: valine [30,31,32,33,34,35,36,37,38,39,40,41,42,43,44,45,46,47,48,49,50,51].

**Table 2 molecules-27-01343-t002:** Biopeptide databases (all web addresses were verified on 28 January 2022).

Database	Web Address	Content	Reference
AHTPDB *	http://crdd.osdd.net/raghava/ahtpdb/	Antihypertensive peptides	[136]
AntiTbPdb	http://webs.iiitd.edu.in/raghava/antitbpdb	Antitubercular and mycobacterial peptides	[137]
APD	http://aps.unmc.edu/AP/main.html	Antimicrobial and anticancer peptides	[138]
AVPdb	http://crdd.osdd.net/servers/avpdb/	Antiviral peptides	[139]
BaAMPs	http://www.baamps.it/	Antimicrobial peptides tested against microbial films	[140]
BactPepDB	http://bactpepdb.rpbs.univ-paris-diderot.fr/cgi-bin/home.pl	Bacterial peptides	[141]
BIOPEP-UWMTM *	http://www.uwm.edu.pl/biochemia	Bioactive peptides/sensory peptides and amino acids	[142]
Brainpeps	http://brainpeps.ugent.be/	Blood–brain barrier passing peptides	[143]
CAMP_R3_	http://www.camp.bicnirrh.res.in/	Antimicrobial peptides	[144]
CancerPPD	http://crdd.osdd.net/raghava/cancerppd/index.php	Anticancer peptides and proteins	[145]
CPPSite 2.0	http://crdd.osdd.net/raghava/cppsite/	Cell-penetrating peptides	[146]
DBAASP	https://dbaasp.org/	Antimicrobial peptides	[147]
EROP-Moscow	http://erop.inbi.ras.ru/	Bioactive peptides	[148]
Hemolytik	http://crdd.osdd.net/raghava/hemolytik/	Hemolytic and non-hemolytic peptides	[149]
MBPDB *	http://mbpdb.nws.oregonstate.edu/	Milk protein-derived bioactive peptides	[150]
NeuroPep	http://isyslab.info/NeuroPep/	Neuropeptides	[151]
PepBank	http://pepbank.mgh.harvard.edu/	Bioactive peptides	[152]
Quorumpeps	http://quorumpeps.ugent.be/	Quorum sensing signaling peptides	[153]
SATPdb	http://crdd.osdd.net/raghava/satpdb/links.php	A metabase of therapeutic peptides	[154]
StraPep	http://isyslab.info/StraPep/	Structures of bioactive peptides	[155]
THPdb	http://crdd.osdd.net/raghava/thpdb/index.html	FDA-approved therapeutic peptides	[137]
TumorHoPe	http://crdd.osdd.net/raghava/tumorhope/	Tumor homing peptides	[156]
YADAMP	http://yadamp.unisa.it/about.aspx	Antimicrobial peptides	[157]

* Databases containing information about peptides derived from food sources were marked with an asterisk.

**Table 3 molecules-27-01343-t003:** Peptide fractions with antioxidant activity.

Source	Characteristics	Procurement and Identification	Activity	Reference
Muscle of miiuy corvina (*Miichthis* *miiuy*)	YASVV, NFWWP, FWKVV, TWKVV, FMPLH, YFLWP, VIAPW, WVWWW, MWKVW and IRWWW	UF/RP-HPLC	Radical scavenging of DPPH and ABTS.	[172]
Finger millet	TSSSLNMAVRGGLTR and STTVGLGISMRSASVR	Trypsin, pepsin MALDI-TOF/TOF–MS/MS	Capture of hydroxyl radicals, DPPH, ABTS and chelating activity.	[173]
Salmon jelly	PP, GF, GPVA, GGPAGPAV, R,Y	Alcalase, Flavourzyme 500 L, Corolase PP, Promod RP-HPLC/UPLC–MS/MS	Inhibitor of dipeptidyl peptidase IV (DPP-IV) and reactive oxygen species (ORAC).	[174]
Shells from shrimp discard processing	SYELPDGQVITIGNER, YPIEHGIITNWDDMEK, EEYDESGPGIVHR, EVDRLEDELVNEK, ALSNAEGEVAALNR, NLNDEIAHQDELINK, LEQTLDELEDSLER	Trypsin, α-chymotrypsin, pepsin GPC NANO LC–LTQ MS	ABTS, DPPH and hydroxyl radical scavenging, reducing power and chelating capacity of ferrous ions, inhibition of β-carotene bleaching, peroxidation of cholesterol and peroxyl and hydroxyl radicals.	[175]
Salmon trimmings	GPAV, VC Y FF	Alcalase, Flavourzyme 500 L, Corolase PP, Promod RP-HPLC/ UPLC–MS/MS	Inhibitor of dipeptidyl peptidase IV (DPP-IV) and reactive oxygen species (ORAC).	[176]
Donkey milk	EWFTFLKEAGQGAKDMWR, GQGAKDMWR, REWFTFLK, MPFLKSPIVPF	MDLC micro-HPLC-Orbitrap–MS	Antioxidant capacity and inhibitory activity of angiotensin-converting enzyme.	[177]
Goat milk	Serum: 883.47–1697.82 Da Casein: 794.44–1956.95 Da Presence of P and H residues.	Pepsin HPLC RP-HPLC	DPPH and superoxide radical scavenging activity.	[178]
Palm kernel	<3 kDa VVG-G-D-G-D-V VPVTST LTTLDSE	Pepsin, pancreatin. UF RP-HPLC	Radical scavenging activity ABTS. Iron-reducing power.	[179]
Cod muscle	MW fractions < 1 kDa	Pepsin, trypsin, chymotrypsin RP-HPLC	Capture of oxygen radicals and DPPH. Superoxide and hydroxyl scavenging activity. Chelating activity of iron. Inhibition of the oxidation of linoleic acid Dose-dependent.	[171]
Juglans sigillata seeds	Peptides with 2–4 aa residues rich in Y, C	Pancreatin GPC RP-HPLC	Radical scavenging activity DPPH, ABTS, oxygen. Iron chelation. Protective activity in PC12 cells against H_2_O_2_-induced cytotoxicity.	[180]
Egg white powder	DHTKE 628.64 Da FFEFH 630.71 Da MPDAHL 684.1 Da	Alcalase UF GPC	628.64 Da. Better oxygen radical absorption capacity. 630.71 and 684.1 Da. DPPH radical scavenging activity. Acidic, hydrophobic and low-molecular weight peptides showed higher antioxidant activity.	[181]
Squid	WCTSVS 682.5 Da	α-chymotrypsin IC GPC	Free radical scavenging activity (DPPH, hydroxyl, superoxide). Chelation of metals (Fe). Avoids DNA damage. Inhibits lipid peroxidation (linoleic acid).	[182]
Campana buffalo mozzarella cheese	CKYVCTCKMS 1326.5 Da	Gastrointestinal digestion in vitro GPC HPLC	Gut protection against induced oxidative stress (CaCo2 cells).	[183]
Bovine hair	CERPTCCEHS 1325.4 Da	Alcalase GPC SEC	ABTS and hydroxyl radical scavenging activity. Inhibition of erythrocyte hemolysis and lipid peroxidation. Protection of DNA and PC12 cells against hydrogen peroxide-induced oxidative damage.	[184]
Plum (*Prunus domestica* L.)	MLPSLPK, HLPLL, NLPLL, HNLPLL, KGVL, HLPLLR, HGVLQ, GLYSPH, LVRVQ, YLSF, DQVPR, LPLLR, VKPVAPF.	High-intensity focused ultrasound. Alcalase RP-HPLC-ESI-Q-TOF	Antioxidant and antihypertensive activity.	[185]
Velvet bean (*Mucuna* *pruriens*)	<1 kDa, 1–3 kDa	Sequential hydrolysis Alcalase–Flavourzyme, pepsin–pancreatin UF	Radical scavenging activity DPPH. Iron-reducing power.	[186]
Peanut seed	TPA (286 kDa) I/LPS (315 kDa) SP (202 kDa)	Alcalase UF GPC HPLC	Peptides with MW < 3 kDa showed greater reducing power than those with PM > 3 kDa.	[187]
Duck egg (egg white)	202.1 294.1 382.1 426.3 514.4 Da	Sequential hydrolysis (alcalase and specific hydrolase for egg protein (SEEP)) GPC IC	Oxygen and hydroxyl radical absorption capacity.	[188]
Palm oil extraction residue	AWFS 509.56 Da WAF 422.48 Da LPWRPATNVF 1200.41 Da	Papain Fractionation based on isoelectric point RP-HPLC	Radical scavenging activity of DPPH. Iron chelation.	[189]
Sweet potato	YYIVS 643.2 Da TYQTF 659.4 Da SGQYFL 713.2 Da YYDPL 669.3 Da	Alcalase UF RP-HPLC	Hydroxyl radical scavenging activity.	[110]
Bovine plasma		Alcalase UF IC RP-HPLC	Free radical scavenging capacity. High reductive power.	[190]

CN: casein; LA: lactoalbumin; LG: lactoglobulin; f: fragment; A: alanine; R: arginine; N: asparagine; D: aspartic acid; C: cysteine; E: glutamic acid; Q: glutamine; G: glycine; H: histidine; Hp: hydroxyproline; I: isoleucine; L: leucine; K: lysine; M: methionine; F: phenylalanine; P: proline; S: serine; T: threonine; W: tryptophan; Y: tyrosine; V: valine; UF: ultrafiltration; RP: reversed-phase; HPLC: high-performance liquid chromatography; MALDI: matrix-assisted laser desorption/ionization; TOF: time of flight; MS: mass spectrometry; UPLC: ultra-performance liquid chromatography; GPC: gel filtration chromatography; LTQ: linear ion trap mass spectrometer; MDLC: multidimensional liquid chromatography; IC: ion-exchange chromatography; SEC: high resolution size exclusion chromatography; ESI: electrospray ionization; Q: quadrupole.

**Table 4 molecules-27-01343-t004:** Relationship between antioxidant activity, amino acids, and mechanism of action.

Amino Acid	Mechanism of Action	Example	Reference
Cysteine	SH groups are radical scavengers, protect tissues from oxidative stress and improve glutathione peroxidase activity.	Tripeptides with C.	[197,199]
Hydrophobic amino acids	Increase solubility of peptides in lipids, facilitating access to hydrophobic radical species and polyunsaturated fatty acids.	P, H or T, within sequences and V or L at N-terminus in peptides. Terminal amino acids such as L or V and G and P residues in gluten peptide sequences.	[77,162,192]
Acidic and basic amino acids	The carboxyl and amino groups of the side chains chelate metal ions due to their ability to dissociate and be proton donors.	A (acidic amino acid) and H (basic amino acid).	[103]
Aromatic amino acids (Y, H, W, and F)	They stabilize radicals by donating electrons, maintaining their own stability via the resonance of their structure.	H at N-terminus. H at C-terminus. Tripeptides with W or Y at C-terminus.	[162]

**Table 5 molecules-27-01343-t005:** Peptide fractions with antimicrobial activity.

Source	Antimicrobial Peptide	Reference
Green coconut water (*Cocos nucifera* L).	CnAMP1, CnAMP2 CnAMP3	[227]
Stems, seeds, and leaves of plants	Thionins Defensins Snakins	[228]
Sardinella prepared by treatment with *Bacillus subtilis* A26 protease	Sardinella protein hydrolysate (SPH),	[226]
Atlantic mackerel (*Scomber scombrus*) Protamex protein hydrolysate	SIFIQRFTT, RKSGDPLGR, AKPGDGAGSGPR GLPGPLGPAGPK	[229]
Protein hydrolysate of anchovy cooking wastewater	GLSRLFTALK	[230]
Frogs	Aurein 1–2 Brevinin 1 Maculatins Citropin	[231,232,233,234,235]
Insects	Cecropin	[236]
Horse Crab	Tachyplesins Polyphemusin	[237,238]
Pigs	Protegrins Tritrpticin	[236]
Bovine	Cathelicidin Indolicidin	[236]
Cow	Bactenecin 1 b-defensin Bac 5 Indolicidin	[238]
Mammals	α defensins β defensins	[239,240]
Honeybee	Apidaecin	[238]
Trypsin digested *Botrytis* *cinerea* protease	FPGSAD, SCVGTDLNR, VAHLT VQGGDAYYLNR SGSTASAVGASLCR	[241]
*Lactococcus lactis, Bacillus subtilis,* and *Bacillus brevis*	Nisin Gramicidin	[242]

## Data Availability

Not applicable.

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
