# Peer review of "Antioxidant and Antimicrobial Peptides Derived from Food Proteins"

_molecules, 2022, doi:10.3390/molecules27041343_

Round 1

Reviewer 1 Report

The manuscript deals with an interesting topic in food science but the scope covered in this work is too general, resulting in significant overlapping with previously published reviews having more or less the same title. Having said that, the novelty of this manuscript is not evident. There is a lack of critical analysis throughout the entire manuscript which, in its present form, appears more of a compilation of data.

Major comments:

The scope (bioactive peptides from food) is too broad. Considering the wide range of food sources, the authors, in many instances,  did not manage to cover most of the literature. Hence, interpretations of the trends observed and conclusions drawn from such observation may not be accurate.

Most of the data come from in vitro work but in vivo data is still lacking. This could be discussed more critically in the manuscript. Section 6 on the bioavailability of the peptides is too brief.

Please check if the examples of antimicrobial peptides in Section 5.2 fulfill the criteria of bioactive peptides (=released after enzymatic analysis and other chemical reactions) and not including "native proteins".

Moreover, interest in employing bioinformatics tools to accelerate research on bioactive peptides can be covered in this manuscript.

Provide a more reasonable concluding remarks. Claims in lines 620-623 are should be substantiated by scientific evidence.

The abstract doesn't truly reflect the content of the manuscript. Topics like problems in the production of bioactive peptides (line 20) and production technologies (line 21) were either not elaborated or discussed too briefly.

Section 3: The authors acknowledged fungi is an important source of bioactive peptides (line 110) but this is not elaborated in subsequent sections.

The compilation of examples of bioactive peptides in Section 3 is not exhaustive. Many important examples are left out.

Section 4: The authors can make a comparison of the advantages and disadvantages of the various emerging technologies.

Lines 312-313: Examples should be highlighted

Table 3 is difficult to read. Suggest to re-organise the information.

The previous work on antimicrobial peptides from food sources should be tabulated.

Minor comments:

The authors are suggested to use the commonly used terminologies, such as oxygen reactive species (ORS) or reactive oxygen species (ROS) (line 33).

Please check and ensure correct chemical formulae are used, for example, nitric oxide and peroxynitrite.

Scientific names in italics.

Please follow the standard in text citations, for example, the sentences in lines 560 and 589.

Author Response

This document contains our reply to comments from reviewer 1 on Manuscript ID: molecules-1555868. We are sure that these changes as well as those made in answer to reviewer’ comments have greatly improved our manuscript, and would like to thank you for their kind help and suggestions.

Thank you so much. We appreciate your time.

Best Regards

Octavio Dublán-García

Laboratorio de Alimentos y Toxicología Ambiental,

Facultad de Química

Universidad Autónoma del Estado de México

Reviewer 2 Report

This review summarized the antioxidant and antimicrobial peptides from food proteins, which can be used as ingredients for functional foods. However, the present manuscript can not be published and need to be major revised.

1 All the introduction introduced the potential harmful effects of oxidative stress on human health and derived that the prevention of the excessive production of free radicals in human body is necessary and useful. Form this point, the introduction gives all readers that the manuscript should be closely related to the antioxidant peptides derived from food protein, but no relation to the antimicrobial peptides. So, the introduction part should be major revised.

2 Peptides from food protein can be produced from enzymatic hydrolysis and microbial fermentation, but chemical synthesis is not suitable. I think chemical synthesis can be deleted.

3 the sentence in lines 352-353 should be revised.

4 the introduction of antimicrobial peptides was not sufficient since even no examples that peptides from food protein were referred. Like antioxidant peptides, a table that summarized the references reported antimicrobial peptides can be given.

5 other activities that benefit the human health were also introduced in the manuscript, like ACE inhibition and so on, which give the sense to readers that the content in manuscript is not related to the title. So, I think the authors should reconsider the title.

Author Response

This document contains our reply to comments from reviewer 2 on Manuscript ID: molecules-1555868. We are sure that these changes as well as those made in answer to reviewer’ comments have greatly improved our manuscript, and would like to thank you for their kind help and suggestions.

Thank you so much. We appreciate your time.

Best Regards

Octavio Dublán-García

Laboratorio de Alimentos y Toxicología Ambiental,

Facultad de Química

Universidad Autónoma del Estado de México

Reviewer 3 Report

Dear Editor,

The current review article performed by López-García et al., 2021 focuses on the antioxidant and antimicrobial peptides derived from food protein, discussing the currently-available technologies for the production, synthesis, sources and bioactive properties. After reviewing the entire manuscript carefully, I found that the work is revealing and can be considered for publication after some major revisions. The detailed suggestion/comments have been given as follows:

Observations

Abstract

The abstract is written clearly, and states briefly the background, scope and approach. However, author should provide some key findings and conclusions of this reviews in order to have a concise and factual abstract.

L15: There is a typo error “degenerative”.

Introduction

L33-34. I would rather to change the words “oxygen reactive species (ORS)” and “nitrogen reactive species (NRS)”, since they might be wrong. They are observed on papers as “reactive oxygen species (ROS)” (as it is even written on the line 58) and “reactive nitrogen species (RNS)”. Please check them.

L60-73. the redaction could improve notably using some periods in order to make shorter and clearer sentences. This remark is valid throughout the manuscript and would help to improve greatly the reading of this manuscript.

L83. Authors should use one period on this line in order to separate the endogenous antioxidant from those the antioxidant obtained from a diet. Improve redaction.

The introduction talks about the free radical production and its importance mainly, which is well described. However, on L 83-87, the authors talk about bioactive peptide and it seems to do not be enough. Authors should include an adequate background of bioactive peptide.

Besides, the introduction is mainly focused on oxidative stress but there is nothing on antimicrobial activity. We have some difficulties to make correlations - if there are some - between both activities, antioxidative and antimicrobial.

Please use italic for works as in vivo, in vitro.

The part 2 intittled “Bioactive peptide” is too short to stand alone. This part should be integrated in the actual part 3 “Sources of bioactive peptides”.

L102. These activities are determined => depend on various parameters among them…

I would merge the part 2 with the small paragraph (108-116) below the actual title 3.

Please correct all the title numeration throughout the manuscript.

Paragraph 3.1. The tittle is on peptides of animal origin. Please focus directly on peptides and nothing else as vitamins…

Throughout their manuscript, the authors discuss other bioactivities than antioxidant and antimicrobial one. They should focus on these 2 bioactivities  solely according to the title of the review. It would be more pertinent and less confusing for the readership.

In table 1.

The format of table 1 should be reworked in order to follow well each line, each activity according to each source of protein origin.

The information provides for hypocholesterolemic shows just the amino acid glycine as protein of origin. Please check it. 

L163-164, table 1 shows some bioactive peptides from food-by products, however, it does not, or it is not clear. Authors should reshape it in order to point out the food protein sources.

L179. The caption of figure 1 should provide the meaning of the RP-HPLC abbreviation and other used.

There is a typo error in Figure 1 (Animal or vegetable protein). The arrow for exogeneous protease is in the bad direction. Panel b) error in synthesis. Caption: production processes

L204. The subtitle microbial fermentation must be 4.2. The same observation for Line 226, 264.

L216: begin with the author et al and not only a reference number. Idem L218.

Line 218-223. The inhibition of free radical and peroxidation is given by peptides produced by acid bacteria, however, when the authors report the references 64 and 65, that inhibition is attributed to the acid bacteria instead of bioactive peptide, which makes confusing the paraph. Improve the redaction.  

L223. The amino acid sequences should be written with the single-letter amino acid code to be uniform through the whole document.

L226. This section should mention that the peptide synthesis is one of the last steps of current approach of food-protein bioactive peptides production, which is often used to validate the biological activities of a discovered peptide, this, in order to make a link with the review topic.

L235. The figure 1 panel b mentioned is not specific for liquid phase peptide synthesis in comparison of solid phase peptide synthesis. In you panel (b) be you should make appear a difference between liquid phase synthesis and solid-phase peptide synthesis.

L264-317. The review informs about the combined use of enzymatic hydrolysis and other emergent technologies such as hydrostatic high-pressure, microwaves, and pulsated electric fields to produce bioactive peptides with less time and lower cost production compared to the usual methods. However, authors should provide a suitable reason why the coupling of the emerging technologies mentioned could actually be a feasible method, knowing that there are going to be more steps included in the process.

L272. Avoid beginning with a number and not the author’s name throughout the text.

L276. specify the duration for the treatment of 500 MPa.

L280. Make a paragraph for microwaves treatment.

L306-310: please cut the sentence, this is too long.

L312: => these treatments exhibited improved antioxidant and functional properties.

L318. 5. Pharmacological properties… and not 5.0

L339: used either as, as or… (sentence to reformulate to be comprehensible)

L341. hydrolysates used as condiments? please give some examples of them.

L352. Grammatic mistake, “lyst” instead “list” of ….

L353, 354. According to this sentence, there are two types of table 2, a) and b), however, it might be wrong, since the letter a and b found here refer to same thing in the Table 2 with the same references (39-41). Please check it.

Table 2. The caption of this table should provide all information about the abbreviations. On the other hand, the information of the column 3 “procurement and identification” is not uniform, since for some of them the information include the proteolytic enzymes, and for other is missed. I would rather to put first, the enzyme treatment (Alcalase, Trypsin, Flavourzyme…etc), follow the purification chromatographic techniques (UF, GFC, RP-HPLC) and finally the identification (MS/MS, MALDI-TOF/TOF, etc). Keep using the single-letter amino acid code through the whole Table 2 and whole document, including tables. Globally, this Table 2 should be reformatted since this is really difficult to read it. Each line should be more separated. The tittle of head of colums should be written in bold.

Figure 2. You should add a caption on each mechanism described such as metal-chelation, radical scavenging, inhibition of lipid peroxidation.

L375. Concerning Arginine, please precise the reference since this is the first time I read something about arginine.

L380. Phe and not Phen.

L394. The word “Aminoácidos” is a Spanish word, please to change it to amino acids. It should be entitled amino-acid or class of amino-acid.

Table 3. the writing in all table is too much. Authors should write in a more synthetic way.

Throughout the manuscript, the sentences are far too long…

L422, authors should provide a reference for this small paragraph.

I must admit that I am not specialized on antimicrobial peptides and cannot review appropriately the second part of this review.

L465. According to the lectine definition given in the manuscript, this would fall outside the scope of this review.

L500. Microorganism.

Paragraph 5.3. I would remove it since this is not the scope of the review, which is dedicated to the antioxidant and antimicrobial peptides.

L554, Keep using the single-letter amino acid code through the whole document

Line 584. It could be interesting to mention some research about the bioavailability of bioactive peptides obtained from food protein hydrolysate in order to have ideas about how it has been carried out and some interesting result found currently.

L611-614. It depends of the initial size of the peptide. This sentence should be moderated.

Conclusion

The authors include the limitations of bioactive peptide production from food protein. Indeed, ideas for future research have been mentioned.

The information obtained in this review will be so useful to know the state of bioactive peptide from food protein nowadays. However, in some points, the authors focus too much on theoretical concepts about peptides instead of comparing experimental results found in bioactive peptide production, which should be one of the most important parts of the review.

Overall, the manuscript is revealing, and deserve acceptance after major revisions.

Author Response

This document contains our reply to comments from reviewer 3 on Manuscript ID: molecules-1555868. We are sure that these changes as well as those made in answer to reviewer’ comments have greatly improved our manuscript, and would like to thank you for their kind help and suggestions.

Thank you so much. We appreciate your time.

Best Regards

Octavio Dublán-García

Laboratorio de Alimentos y Toxicología Ambiental,

Facultad de Química

Universidad Autónoma del Estado de México

Round 2

Reviewer 1 Report

This review manuscript, in my opinion, is not considered very comprehensive, considering the fact that bioactive peptides is indeed a very broad topic. Nonetheless, revision made by the authors based on the reviewers comments has improved the scientific quality of this review.

I do not have further comments on the scientific content but I feel that this manuscript may benefit from language correction by a native English speaker. In addition, I noticed some typographical errors that can be corrected.

  1. Line 125: Table 1
  2. Lines 187-188 and other parts: Not advisable to have a paragraph with only one sentence?
  3. Line 397: GRAS should be defined when it is first mentioned.
  4. Line 747: Figure. 3.
  5. Lines 80-804: use small letters for zebrafish
  6. Be consistent when using Gram-positive or gram positive

Author Response

February 10th, 2022

Response to reviewer 1

Molecules-1555868

Title: “Antioxidant and antimicrobial peptides derived from food proteins

This document contains our reply to comments from reviewer 1 on Manuscript ID: molecules-1555868. We are sure that these changes as well as those made in answer to reviewer’ comments have greatly improved our manuscript, and would like to thank you for their kind help and suggestions.

Reviewer 1

This review manuscript, in my opinion, is not considered very comprehensive, considering the fact that bioactive peptides is indeed a very broad topic. Nonetheless, revision made by the authors based on the reviewers comments has improved the scientific quality of this review.

I do not have further comments on the scientific content but I feel that this manuscript may benefit from language correction by a native English speaker. In addition, I noticed some typographical errors that can be corrected.

  1. Line 125: Table 1

Reply: Thank you so much for your comment, the error has been corrected.

  1. Lines 187-188 and other parts: Not advisable to have a paragraph with only one sentence?

Reply: Thank you for your comment, the error was corrected throughout the manuscript.

  1. Line 397: GRAS should be defined when it is first mentioned.

Reply: Thank you so much for your comment, the error has been corrected.

  1. Line 747: Figure. 3.

Reply: Thank you so much for your comment, the error has been corrected.

  1. Lines 80-804: use small letters for zebrafish

Reply: Thank you for your comment, the error was corrected throughout the manuscript.

  1. Be consistent when using Gram-positive or gram positive

Reply: Thank you for your comment, the error was corrected throughout the manuscript.

Thank you so much. We appreciate your time.

Best Regards

Octavio Dublán-García

Laboratorio de Alimentos y Toxicología Ambiental,

Facultad de Química

Universidad Autónoma del Estado de México

Reviewer 2 Report

the manuscript has been well revised and organized that can be accepted for publication.

Author Response

February 10th, 2022

Response to reviewer 2

Molecules-1555868

Title: “Antioxidant and antimicrobial peptides derived from food proteins

We appreciate all your comments and observations you made to the manuscript, with this, it improved significantly.

Thank you so much. We appreciate your time.

Best Regards

Octavio Dublán-García

Laboratorio de Alimentos y Toxicología Ambiental,

Facultad de Química

Universidad Autónoma del Estado de México
